# The Importance of Rumors in the Spanish Sports Press: An Analysis of News about Signings Appearing in the Newspapers Marca, As, Mundo Deportivo and Sport

**Francisco-Javier Herrero-Gutiérrez [1,\*] and José-David Urchaga-Litago [2]**

[1] Department of Sociology and Communication, University of Salamanca, 37008 Salamanca, Spain
[2] Department of Communication, Pontifical University of Salamanca, 37002 Salamanca, Spain; jdurchagali@upsa.es
\* Correspondence: javiherrero82@usal.es; Tel.: +34-923-294-500 (ext. 3139)

**Abstract:** The front pages of newspapers are the main showcase to sell the product. Those first pages are a perfect hook for newspapers to attract readers; thus, it becomes vital to show striking pieces of information, captivating the audience. In the case of the written sport press in Spain, there is a key period in which true information is mingled with half-truths and even rumors: The summer transfer window. This paper shows an analysis of the front-page news appearing in the Spanish sports newspapers Marca, As, Mundo Deportivo, and Sport, over a five-year period (2015–2019), based on a sample of 120 different issues of the newspaper. Many times, the media present information either as something true or as a hypothesis or possibility. After quantitatively analyzing that, it can be noticed that in more than 50% of the cases, the signing or sale of the player referenced on the front page (the main news) does not occur. Similarly, it can be observed that there is a direct link connecting the news referring to Real Madrid with Marca and As, and Fútbol Club Barcelona with Mundo Deportivo and Sport. Finally, almost 100% of this news is showed along with real photographs, using photo montage in just a few cases.

**Keywords:** sport press; Marca; As; Mundo Deportivo; sport; rumor; signing; transfer; Real Madrid; Barcelona; journalistic rumor

## 1. Introduction

Nowadays, Marca—a sports newspaper—stands for the most widely read newspaper in Spain, representing more readers than any other general-interest newspaper, such as El País or El Mundo. That is according to results that have arisen from the General Media Study (Estudio General de Medios—EGM) [1], released on a quarterly basis by the Spanish Association for Media Research (Asociación para la Investigación de los Medios de Comunicación—AIMC).

Then, it is no mere anecdote that, according to the AIMC, amongst the 10 most-read newspapers in Spain, there are four specialized in sports news. Apart from Marca, we have to mention As, Mundo Deportivo, and Sport, also in the top 10 (Figure 1).

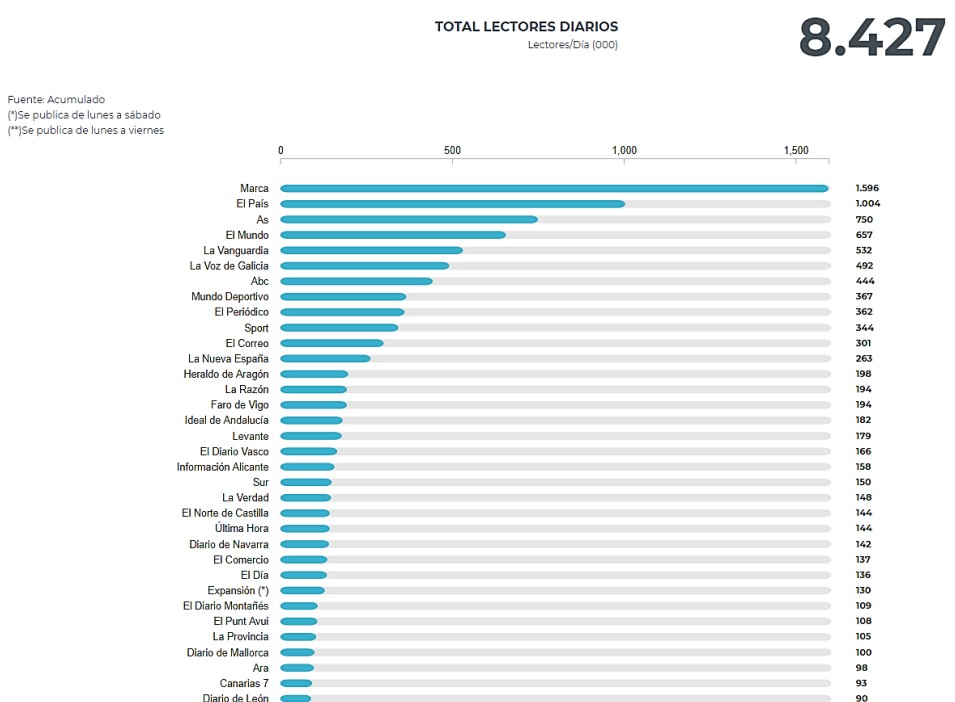

**Figure 1.** Ranking of the most read newspapers in Spain according to the Estudio General de Medios (EGM) carried out by the Asociación para la Investigación de los Medios de Comunicación (AIMC). Source: EGM 2020. Source: https://reporting.aimc.es/index.html#/main/diarios.

It is true that there is a declining trend in the number of readers, as it is the case of the rest of in-print newspapers in this country, and virtually worldwide. This situation is similar for both the written press (newspapers) and magazines or newspapers supplements (Figure 2).

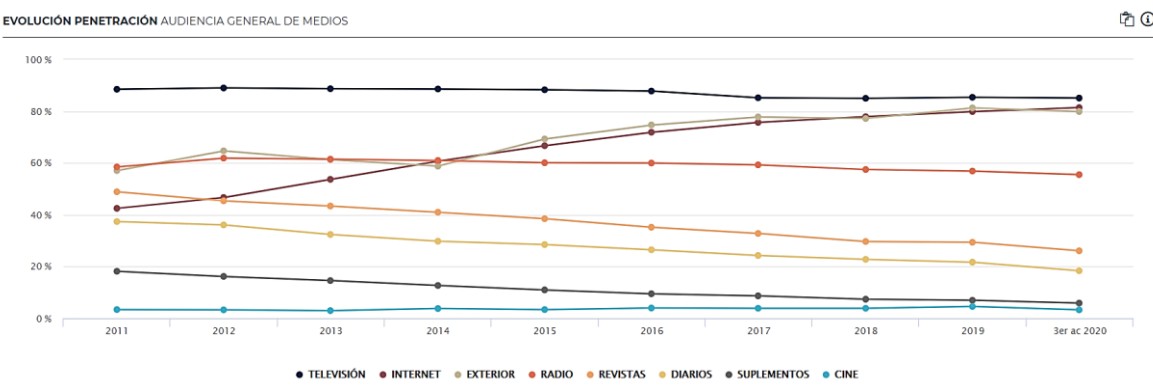

**Figure 2.** In-print media trend over the last decade according to the EGM. Source: EGM 2020. Source: https://reporting.aimc.es/index.html#/main/cockpit.

In terms of media impact, Spaniards consume far more sport news than in any other country, applying the classic division between sport-practice and sport-entertainment [2] (p. 4). This is verifiable. To do so, we could take any media as an example.

Apart from the one already mentioned, that of the written press, we cannot ignore the fact that the most watched TV programs in Spain are related to sports. Amongst them, sports broadcasts are in the lead, according to different companies specialized in audience measuring, such as the Kantar group.

For example, on the Formulatv.com website [3], which draws on data from Kantar Media, taking as a reference the last three years, in 2020, 3 of the 10 most watched television

events of the year (the first 3) were sporting events [4]; in 2019, 6 of the top 10 most viewed events of the year were sporting events [5]; and in 2018, the top 10 most viewed television events were sports content [6].

Moreover, it is noted that the main generalist radio broadcasters in Spain, including the four most listened (Cadena SER, COPE, Onda Cero, and Radio Nacional de España), devote a lot of their time to sport issues, especially during the weekends. Those last are seized by sport events, especially association football (henceforth, football) broadcasting. Besides, we cannot forget to mention Radio Marca, exclusively focused on sport news.

It is common for mass media to analyze the most popular shows after each sporting event. Among the 10 most followed, from Monday to Friday, there are almost always night sports programs such as "El Larguero" (SER channel) or "El Partidazo" (COPE); and broadcast shows are the most listened to on weekends [7].

Nor should we ignore the importance of websites such as marca.com [8], as.com [9], mundodeportivo.com [10], or sport.es [11]. Some of them are supported by the number of users whose data have been certified by comscore.com or introl.es, a subdivision of the interactive Spanish Broadcast Verification Office (Oficina de Justificación de la Difusión—OJD) [12]. Along with all this, we have social media, where sport news is shared referencing the previous media and others [13–15]. In conclusion, the importance of sport-entertainment within Spain has been clearly demonstrated [16,17].

Usually, when we refer to all these mass media, including the written press (under consideration on this paper) we can divide sports information into two large groups: Sport broadcasts, that is, live information; and news (news, articles, reports, interviews), that is, on-demand information.

When dealing with live information (especially broadcasts), it can rarely be conveyed as rumors or false news, nowadays encompassed in the term fake news. If a football game is broadcasted, the result will be immovable and hence, difficult to fake. A certain team will win and that is irrefutable. There is a different issue regarding the greater or lesser subjectivity with which said event is broadcasted, influenced by multiple factors. As an example of that, we can find the mere placing of opinion remarks next to the factual information, and even the opinion itself disguised as information, thus trying to share as factual information something that, in fact, is not. However, we would find ourselves surrounded by subjectivity and shifting words, since it becomes especially difficult to measure statements such as "a team played well/badly" or "such an athlete deserved/did not deserve to win".

It is in the second group that we mentioned where we can face information that, in fact, is not such. Within this second group, we encounter a wide range of journalistic genres (report, interview, news, articles . . . ); amongst them, there is perhaps one that stands out in terms of rumors contained: The news. It is precisely that which is intended to be carried out and reflected on this paper, focusing on the front pages of newspapers and the news showed in them. We will limit the sample to news about possible player signings or sales carried out during a specific period, which will be later detailed.

## 2. Theoretical Framework

Founded in Barcelona in 1856, El Cazador can be considered as the first sports publication in Spain [18–20]. Since this very first publication appeared, many others have emerged in the field, showing different particularities especially on periodicity and changing lifecycles.

These days, as far as sports press is concerned—not taking into account specialized magazines but only daily published press with varied sports information—there are four widely read sports newspapers; as already indicated in the introduction: Marca, As, Mundo Deportivo, and Sport, with Marca in the lead.

Currently based in Madrid, Marca was founded in San Sebastián as a weekly newspaper in 1938, amidst the Spanish Civil War. Currently, its main headquarters are in Madrid. In terms of information, its most requested one is that related to Madrid teams, especially

Real Madrid. As dates from 1967 and it is also based in Madrid. Its target is focused on Madrid as well, especially in its two main teams: Real Madrid and Atlético de Madrid.

First published in 1906, Mundo Deportivo has its headquarters in Barcelona. It is one of the 10 most widely read Spanish newspapers and its target group comprises mainly FC Barcelona fans. Founded in 1979, also established in Catalonia, and a pioneer in the use of color in its pages, Sport also bases its information on FC Barcelona.

The four mentioned sports newspapers offer information on all Spanish clubs, with football being their main topic (between 50% and 80% of the information on their pages, depending on the period of the year and the newspaper).

### 2.1. The Importance of the Front Page

Obviously, as it is the case of all printed media, the front page is an ideal showcase, the first thing the reader sees, an "open window" [19] (p. 89). We will focus the study on front pages because "within the agenda settings established by the media, in this case by the newspapers, the preferential news will take large spaces on the front page and, therefore, will be considered as the main news" [21] (p. 1355).

The front page is the first contact with the reader [22] (p. 145). Therefore, "it is important to amaze and inform, using for that 'content inserted into a container'" [19] (p. 89). In addition, "there are great differences between the front pages appearing on the general information press and those of sports press" [21] (p. 1355). Similarly, when using headlines or photographs: "The design of general information newspapers tend to be somehow visually organized, whereas the sports press uses more daring and silhouetted images on many occasions" [23]. Moreover, we cannot forget the importance of the linguistic flow [24].

However, anything goes when talking about information? We must consider the importance of newspapers sales [25]. Those sales will be based on the amount of striking information showed, especially on the front page—frequently considered as collectible products. It is precisely there where the thin red line between information, rumor, and semi-truths comes into play. Does anything go? Which of the information is true on those front pages? To answer those questions will be the object of this study.

When flicking though a newspaper, the reader "will expect information to be displayed in the usual way of his often-read newspaper, from the very front page to the end. A good, witty cover that you like [...]" [25] (p. 471). However, this is where another question arises: Do newspapers show real information or, on the contrary, just information that the reader would like to be real?

We could pose many questions related to this issue; nevertheless, we do not intend to embrace all of them here. To do so, we would require methodological techniques different from those used to conduct this study. It is difficult to identify what is being told in the newspapers and its intention. Likewise, it is difficult to determine the number of sources of information, their credibility, and whether there has been a cross-checking process. To clarify those aspects, we could interview newspaper editors and journalists. Neither is it easy to measure the satisfaction rate of readers with the information received; however, it could be useful to go further on this, conducting surveys or setting focus groups. Here, we introduce potential lines of research that could be designed considering future data.

The objective of this work is to discover to what extent the news appearing during the summer transfer window is true.

### 2.2. The Thin Line between Rumor, Semi-Truths and Fake News

In the field of journalism, information is clearly differentiated from opinion. The news is information, and it must be based on cross-checked facts (coming from a plurality of sources); thus, it is objective. On the contrary, opinion is subjective. Any journalistic content perceived by the reader as news should be objective and cross-checked [26]. That said, in the sports press, it is quite common to find headlines announcing possible signings that are later not materialized. In those cases, the media justify themselves saying that

there existed rumors; or even, that there was an intention to effect those signings but it did not happen, due to different circumstances. In this case, they argue that the headline did have a true content, but that it could not be entirely realized—which would be a semi-truth. Nevertheless, there is also a third possibility: It was fake news, referring to false information—which may include a part of real content—and has an intention to harm or/and clearly benefit a specific group. There is a thin line between rumor, semi-truths, and fake news, leaving the reader unprotected and journalism called into question.

In order to understand the concept of fake news in any of these variants, research is based on the fact that, almost always, these news are given as true even though they are not, nor will they ever be, beyond the fact that there could be fond indications of veracity which may occur (for example, a transfer), either with a greater or lesser probability. The problem here is the lack of use of the conditional tense, which would surely detract from the ultimate goal of the headline. So it is preferable to start for example "Luis Suárez will sign for Atlético de Madrid" instead of "Luis Suárez could sign for Atlético de Madrid". Obviously, the use of the conditional indicates that "could" or "could not" and the news would cease to be such, when all options were left open.

These three concepts (rumor, semi-truth, and fake news) share a common feature: cContent is not cross-checked. In the case of fake news, they are based on false or misleading information: The reason that they are included within the term misinformation [27–30]. According to Merriam Webster Dictionary (2020), rumor is a "talk or opinion widely disseminated with no discernible source". From this definition two conclusions can be derived. On the one hand, there exists a message; however, we do not know to what extent it is true. On the other hand, there is a mention to a discernible source, who/which we do not know. In his research project, Mazo [31] (p. 47) concludes that rumor in the social communication field is "spontaneous, elusive and spreads exponentially, developing an interesting and ambiguous content which is modified in a metamorphic process; all that with the intention of appearing credible to their audience".

This author explains that rumor possesses some unique characteristics. Its sources are anonymous, but paradoxically they are perceived as very credible. The message is very attractive, appealing, and can create uncertainty, pose questions, and provide impartial answers—for which we need more information. In many cases, it mostly refers to confidential information, the reason that the reader considers it to be something valuable and interesting [32] that has not been officially published yet. Moreover, it also tends to be perceived as true. Telling this information is considered an "imperative need", the reason that it is spread (p. 47). It is shared with a selected and like-minded audience, considered to be interested in the subject and willing to embrace it. The spread of rumor is directly proportional to the interest in the subject and its ambiguity [33].

In the case of sports press readers, rumors about signings are very appealing, since the entering/exiting of players in a team directly affects its performance on the field. Due to these particularities, rumors spread faster than cross-checked news. This is because they are surrounded by some uncertainty, which allows people to take positions for or against. Sharing this type of content reinforces the sense of belonging to a group [31] (pp. 40–41). In the case of sport, this has been utilized to establish ideological links, thus supporting the construction of identities, which are even in some cases ethnic and territorial [34]. In Spain, a country where great internal nationalist political positions and territorial disputes are ubiquitous, football is not alien to this situation. That is the reason that certain teams are usually considered as supporters of certain political views (for example, FC Barcelona regarding the Catalan independence movement, and Real Madrid as defender of the unity of Spain), a situation reflected in the bias of the different media [35].

Hoax is something accepted or established by fraud or fabrication (Merriam Webster, 2020) [26], so it can be considered a synonym for fake news, with the difference that this term is only used by mass media. Shudsonand Zelizer [36] reminds us that the intentional spreading of false news seeking a particular purpose has been on scene since the very beginning of journalism. Thus, they mention how in the USA, Thomas Jefferson told a

friend in 1807: "the man who never looks into a newspaper is better informed than he who reads them; inasmuch as he who knows nothing is nearer to truth than he whose mind is filled with falsehoods & errors".

Lazer has coordinated a group of 16 researchers who have studied the media publishing fake news, proposing this definition: "We define "fake news" to be fabricated in formation that mimics news media content in form but not in organizational process or intent". These media lack of processes and standards to guarantee the truthfulness of their content. In this context, they call for interdisciplinary research to be carried out in order to reduce the spread of fake news, since professionals in journalism must provide objective and reliable information [37] (p. 1094).

### 3. Objectives and Hypotheses

The main objective of this study is to determine to what extent the news appearing on the front pages of newspapers during the summer transfer window (2015–2019) were true; therefore, focusing on a specific type of news: signings and sales of players. It was our decision to not include the year 2020 in our study, since it does not represent an average year for the signing/transfer market (hardly any), due to the COVID-19 pandemic.

The baseline scenarios (hypotheses) are:

1. At least 50% of the signings, possible signings, renewals, or players leaving the teams, announced on the front pages, are not materialized.
2. More than 25% of the signings, possible signings, renewals, or players leaving the teams, announced on the covers, are accompanied by photo montages. In those, the players wear the shirt of the hypothetical team in which they would play.
3. More than 80% of Marca and As front pages are devoted to Real Madrid players, either as a source or as a target.
4. More than 80% of Mundo Deportivo and Sport front pages are devoted to Barcelona players, either as a source or as a target.
5. More than 80% of this news is accompanied by photographs.

### 4. Methodology

The methodology appearing on this paper exclusively responds to quantitative techniques. A series of variables was established (Table 1), in order to resolve the previously formulated hypotheses. It has been deemed appropriate to use this methodology as it includes "a set of, increasingly perfect and constantly improving, methodological instruments applied to extremely diversified forms of speech (both content and container)" [38] (p. 7).

**Table 1.** Variables to analyze. Compiled by author.

| Newspaper | Marca/As/Mundo Deportivo/Sport |
|---|---|
| Date | According to criteria stated in "population and sample" |
| Piece of news | Indicate the piece of news |
| Headline | Indicate the main headline on the front page |
| Name of the player | Indicate the name of the player, subject of the front page |
| Team involved | Indicate name of the team, subject of the front page |
| The piece of news is materialized | Indicate whether the information referred in the piece of news is materialized |
| Includes photograph(s) | Indicate if it includes a photograph of the player referred. |
| Includes photo montage | Indicate if it includes a photo montage featuring the player referred |

Here we face a content analysis, defined by Krippendorff [39] (p. 21) as: "[...] a research technique for making replicable and valid inferences from data to their context". Likewise,

Berelson [40] (p. 18) defines it as "a research technique for the objective, systematic and quantitative description of the manifest content of communication", in the same line as other authors [41].

After conducting a systematic research through the main bibliographic reference sources (mainly WOS and Scopus), as well as recent review papers [42–45], we do not know of any other research papers similar to the analysis proposed here, with content methodology of a representative sample of news items that appeared in the press, in which the researchers themselves contrasted the truth of the news, with the exception of an isolated case taken to the United States sports arena [46]. Therefore, this methodological approach is novel and can be applied to any type of news, not only in the sports field. However, we can find approximations in studies related to fake news in digital media as blogs or online newspapers [47–49], but our study addresses the traditional press where the computerized possibilities of detecting fake news are not so many.

The final veracity of the news was contrasted with knowing if a certain player was still part of the squad or not, through the official data from La Liga (https://www.laliga.com/) [50] taking into account that all players who appear on this headline would have a relationship with the Spanish League (either because they were signing for a Spanish club or were no longer part of the squad of a Spanish club).

### 4.1. Scope of Study

The scope of this study extends only to news about football signings or possible signings analyzed according to certain factors previously detailed.

### 4.2. Population and Sample

The population would be made up of Marca, As, Mundo Deportivo, and Sport copies published in a period of five years (2015–2019) (Table 2).

**Table 2.** Sample. Compiled by author.

|  |  | MARCA | AS | MUNDO DEPORTIVO | SPORT | TOTAL |
|---|---|---|---|---|---|---|
| **2015** | JUL | 3 | 3 | 3 | 3 | 12 |
|  | AUG | 3 | 3 | 3 | 3 | 12 |
| **2016** | JUL | 3 | 3 | 3 | 3 | 12 |
|  | AUG | 3 | 3 | 3 | 3 | 12 |
| **2017** | JUL | 3 | 3 | 3 | 3 | 12 |
|  | AUG | 3 | 3 | 3 | 3 | 12 |
| **2018** | JUL | 3 | 3 | 3 | 3 | 12 |
|  | AUG | 3 | 3 | 3 | 3 | 12 |
| **2019** | JUL | 3 | 3 | 3 | 3 | 12 |
|  | AUG | 3 | 3 | 3 | 3 | 12 |
| **TOTAL** |  | 30 | 30 | 30 | 30 | 120 |

The convenience sample will be delimited according to these criteria:

(A)  Firstly, we will only consider front pages published over the months of July and August, thus coinciding with summer transfer window period in the Spanish football league.

(B)  We will only use the front pages published on the three first days of July; consequently, the first three front pages. Those shall be related to a signing, possible signing, renewal, or a player leaving (cross-checked or hypothetical), starting from July 1.

(C)  Regarding August, the same procedure (indicated in subparagraph b) will be applied.

(D)  We will not take into account front pages dealing on the news event. Just the first one will be considered.

The convenience sample was delimited according to the criteria set by Riffe, Lacy, and Fico: "The material being studied must be difficult to obtain [ . . . ] Resources limit the ability to generate a random sample of the population [ . . . ] The third condition justifying convenient sampling is when a researcher is exploring some underresearched but important task" [51] (p. 85).

## 5. Results

The total number of front pages analyzed, following the criteria described above, was 104. It should be noted that the expected total (120) does not correspond to the actual total. That is so because, in at least in three days each month, the front pages did not coincide with the object of this study, applying the above-established sampling criteria. The selected were the following (Table 3):

**Table 3.** Days chosen to be part of the sample. Compiled by author.

| Newspaper | 2015 | | 2016 | | 2017 | | 2018 | | 2019 | | Total |
| | JUL | AGO | JUL | AGO | JUL | AGO | JUL | AGO | JUL | AGO | |
|---|---|---|---|---|---|---|---|---|---|---|---|
| **Marca** | 3, 7, 15 | 17, 20, 22 | 5, 16 | | 5, 6, 13 | 3, 12 | 3, 14, 18 | 28 | 14, 16, 17 | 2, 7, 9 | 23 |
| **As** | 2, 7, 16 | 2 y 3 | 16, 18, 20 | 7 | 2, 4, 6 | 3, 10, 11 | 6 y 26 | 8, 23 y 29 | 2 y 22 | 2, 5, 9 | 25 |
| **Mundo Deportivo** | 2, 22, 28 | 7 | 1, 2 y 3 | 1, 2 y 13 | 2, 3, 5 | 1, 2, 13 | 5, 6, 8 | 2, 3, 9 | 2, 3, 19 | 1, 7, 9 | 28 |
| **Sport** | 1, 2, 12 | 1, 2, 25 | 1, 2 y 5 | 8, 13 y 16 | 2, 3, 6 | 1, 2, 6 | 3, 4, 6 | 3, 8, 20 | 1, 3, 4 | 10 | 28 |

The first matter analyzed was focused on estimating how many pieces of news (considering hypothetical and assured ones, by newspapers) were, in the end, materialized. The results show that in 56.7% of the cases, those announced events did not happen, so it was just fulfilled by just 43.3% of cases. If we focus our analysis on different newspapers, we note that the most reliable one is Mundo Deportivo, with half of their information, in the end, materialized; however, another 50% is not. Marca and As did not comply with the proclaimed news by 61% and 60% of the cases, respectively, and Sport by 57.1%. The observed differences cannot be considered significant ($\chi^2$ = 0.788; $p$ = 0.852) (Table 4). Some of the examples related to this are (Table 5):

**Table 4.** Compliance rate depending upon newspaper. Compiled by author.

| Newspaper | Compliance with the Piece of News | | | | Total |
| | Yes | | No (Fake News) | | |
| | *n* | % | *n* | % | |
|---|---|---|---|---|---|
| Marca | 9 | 39.1% | 14 | 60.9% | 23 |
| As | 10 | 40.0% | 15 | 60.0% | 25 |
| Sport | 12 | 42.9% | 16 | 57.1% | 28 |
| Mundo Deportivo | 14 | 50.0% | 14 | 50.0% | 28 |
| Total | 45 | 43.3% | 59 | 56.7% | 104 |

**Table 5.** Headline examples. Compiled by author.

| | Headline | Translation into English: | URL Portada |
|---|---|---|---|
| | | **Marca** | |
| André Gomes signing | André Gomes es el tapado | André Gomes, the ace in the hole | https://es.kiosko.net/es/2016-07-05/np/marca.html |
| Di María signing | Di María a tiro del Barça | Barça. Di Maria in range | https://es.kiosko.net/es/2017-08-12/np/marca.html |
| James signing (Atlético del Madrid) | Bombazo a la vista ¡James quiere jugar en el Atlético! | Bombshell. James wants to play in Atletico! | https://es.kiosko.net/es/2019-07-14/np/marca.html |
| Mbappé signing | Espéranos, Mbappé | Wait for us, Mbappé! | https://es.kiosko.net/es/2017-07-06/np/marca.html |
| Neymar signing | Never never never | Never never never | https://es.kiosko.net/es/2018-07-14/np/marca.html |
| Neymar signing | Neymar en el horizonte | Neymar ahoy! | https://es.kiosko.net/es/2019-08-09/np/marca.html |
| Pogbá signing | El United no afloja con Pogba | United doesn't let up on Pogba | https://es.kiosko.net/es/2019-07-17/np/marca.html |
| Pogbá signing | Erikesen podría ser la llave de Pogba | Erikesen could be the key for Pogba | https://es.kiosko.net/es/2019-08-07/np/marca.html |
| Van de Beek signing | Se reactiva la opción Van de Beek | Van de Beek option reactivated | https://es.kiosko.net/es/2019-08-02/np/marca.html |
| De Gea arrival to Real Madrid | Blindado De Gea | De Gea: armored | https://es.kiosko.net/es/2015-07-15/np/marca.html |
| Sergio Ramos leaves Real Madrid | No da marcha atrás | No backtrack | https://es.kiosko.net/es/2015-07-03/np/marca.html |
| Asensio leaving | Hola Kovacic, adiós Asensio | Hi Kovacic, bye Asensio | https://es.kiosko.net/es/2015-08-20/np/marca.html |
| Bale leaving | Bale tiene una salida | Bale has a way out | https://es.kiosko.net/es/2019-07-16/np/marca.html |
| James leaving | La Premier tienta a James | Premier League wants James | https://es.kiosko.net/es/2016-07-16/np/marca.html |
| | | **As** | |
| André Gomes signing | Andre Gomés espera al Madrid | Andre Gomes awaits Real Madrid | https://es.kiosko.net/es/2016-07-20/np/as.html |
| Cavani signing | Cavai se deja querer | Cavai lets himself be pampered | https://es.kiosko.net/es/2018-07-26/np/as.html |
| De Gea signing | El United pone precio: De Gea 35 M | United puts a price for De Gea: 35m | https://es.kiosko.net/es/2015-07-02/np/as.html |
| Gabriel Jesús signing | Gabriel Jesús más cerca del Real Madrid | Gabriel Jesus closer to Real Madrid | https://es.kiosko.net/es/2016-07-16/np/as.html |
| Mbappé signing | Mbappé aún es posible | Mbappé. Still possible | https://es.kiosko.net/es/2018-08-23/np/as.html |
| Pogbá signing | El Madrid pide paciencia a Pogbá | Madrid asks Pogbá for patience | https://es.kiosko.net/es/2016-07-18/np/as.html |
| Pogba signing | Pogba cuenta atrás | Countdown: Pogba | https://es.kiosko.net/es/2019-08-05/np/as.html |
| Van de Beek signing | Plan Van de Beek | Van de Beek operation | https://es.kiosko.net/es/2019-08-02/np/as.html |

**Table 5.** *Cont*.

| | Headline | Translation into English: | URL Portada |
|---|---|---|---|
| Meunier signing | Opción Meunier | Meunier, an option | https://es.kiosko.net/es/20 17-08-11/np/as.html |
| Isco leaving | El Milán quiere a Isco | Milan wants Isco | https://es.kiosko.net/es/20 16-08-07/np/as.html |
| Bale leaving | Gareth sale | Gareth leaves | https://es.kiosko.net/es/20 19-07-22/np/as.html |
| Benzemá leaving | El Arsenal quiere a Benzemá | Arsenal wants Benzemá | https://es.kiosko.net/es/20 15-08-02/np/as.html |
| Neymar leaves PSG | Neymar a subasta | Neymar to auction | https://es.kiosko.net/es/20 19-08-09/np/as.html |
| Ramos leaving | La semana clave de Ramos | Key week for Ramos | https://es.kiosko.net/es/20 15-08-03/np/as.html |
| **Mundo Deportivo** | | | |
| Alaba signing | Alaba, el tapado | Alaba, the ace in the hole | https://es.kiosko.net/es/20 19-07-19/np/ mundodeportivo.html |
| Ceballos signing | Operación Ceballos | Operation Ceballos | https://es.kiosko.net/es/20 17-07-02/np/ mundodeportivo.html |
| Gabriel Jesús signing | Último intento por Gabriel Jesús | Last shot for Gabriel Jesus | https://es.kiosko.net/es/20 16-08-02/np/ mundodeportivo.html |
| Gignac signing | Alternativa Gignac | Alternative Gignac | https://es.kiosko.net/es/20 16-08-01/np/ mundodeportivo.html |
| Neymar signing | Un truque de 170 "kilos" | 170m all-in | https://es.kiosko.net/es/20 19-07-03/np/ mundodeportivo.html |
| Barcelona signs Neymar | Neymar prioridad Barça | Neymar prioritizes Barça | https://es.kiosko.net/es/20 19-08-09/np/ mundodeportivo.html |
| Nolito signing | Oferta por Nolito | Offer for Nolito | https://es.kiosko.net/es/20 15-08-07/np/ mundodeportivo.html |
| Pogbá signing | Pogbá, el deseado | Pogba, the desired | https://es.kiosko.net/es/20 18-08-02/np/ mundodeportivo.html |
| Sciglio signing | Operación carrilero | Operation carrilero | https://es.kiosko.net/es/20 16-07-03/np/ mundodeportivo.html |
| Verratti signing | Cambio de táctica | Shift in tactics | https://es.kiosko.net/es/20 17-07-03/np/ mundodeportivo.html |
| William and Rabiot signings | William y Rabiot en cabeza | William and Rabiot in the lead | https://es.kiosko.net/es/20 18-07-08/np/ mundodeportivo.html |
| Marca de Samper | Wenger ataca de nuevo | Wenger attacks again | https://es.kiosko.net/es/20 15-07-22/np/ mundodeportivo.html |

**Table 5.** *Cont.*

|  | Headline | Translation into English: | URL Portada |
|---|---|---|---|
| Coutinho to Tottenham | Negocian por Coutinho | Coutinho: in negociations | https://es.kiosko.net/es/2019-08-07/np/mundodeportivo.html |
| Rakitic leaving | Ofertas por Rakitic | Offers for Rakitic | https://es.kiosko.net/es/2018-08-09/np/mundodeportivo.html |
| Sport | | | |
| Neymar continues | ¡Bloqueado! | Blocked! | https://es.kiosko.net/es/2017-08-02/np/sport.html |
| Ceballos signing | Ofensiva por Ceballos | Fighting for Ceballos | https://es.kiosko.net/es/2017-07-02/np/sport.html |
| Gabriel Jesús signing | Guerra Madrid-Barça por Gabriel Jesús | Madrid-Barça war for Gabriel Jesús | https://es.kiosko.net/es/2016-07-02/np/sport.html |
| Griezmann signing | Griezmann es el favorito | Griezmann favorite | https://es.kiosko.net/es/2017-08-01/np/sport.html |
| Luan signing | Cuenta atrás por Luan | Countdown for Luan | https://es.kiosko.net/es/2016-08-08/np/sport.html |
| Lucas Pérez signing | Atención a Lucas Pérez | Lucas Pérez! | https://es.kiosko.net/es/2016-07-05/np/sport.html |
| Barcelona signs Neymar | Reunión Messi-Neymar | Messi and Neymar meet | https://es.kiosko.net/es/2019-07-04/np/sport.html |
| Real Madrid signs Neymar | Superoferta de Florentino a Neymar | Florentino superbid for Neymar | https://es.kiosko.net/es/2019-08-10/np/sport.html |
| Paulinho signing | 50 millones por Paulinho | 50m for Paulinho | https://es.kiosko.net/es/2018-07-06/np/sport.html |
| Pogba signing | Las exigencias de Pogba | Pogba demands | https://es.kiosko.net/es/2015-07-01/np/sport.html |
| Verratti signing | Verratti Día D | Day D: Verrati | https://es.kiosko.net/es/2017-07-03/np/sport.html |
| Yarmolenko signing | Yarmolenko opción para enero | Yarmolenko. January option | https://es.kiosko.net/es/2015-08-25/np/sport.html |
| Abdennour signing | Habrá fichaje | Signing to happen | https://es.kiosko.net/es/2015-08-02/np/sport.html |
| Gerson signing | ¡Peligra Gerson! | Gerson at stake! | https://es.kiosko.net/es/2015-08-01/np/sport.html |
| Coutinho to PSG | Acuerdo Coutinho-PSG en la operación Neymar | Operation Neymar: Coutinho-PSG agreement | https://es.kiosko.net/es/2019-07-03/np/sport.html |
| Rakitic and Busquets leaving | Ofensiva del PSG por Rakitic y Busquets | PSG battles for Rakitic and Busquets | https://es.kiosko.net/es/2018-08-20/np/sport.html |

As far as photo montages are concerned, they have only been found in 2.9% of the cases. As and Sport did not use any, whereas Marca published two and Mundo Deportivo one. The observed differences cannot be considered significant ($\chi^2$ = 4.39; *p* = 0.222) (Table 6).

**Table 6.** Use of photo montage depending upon newspaper. Compiled by author.

| Newspaper | Photo Montage | | | |
| | Yes | | No | |
| | n | % | n | % |
| --- | --- | --- | --- | --- |
| Marca | 2 | 8.7% | 21 | 91.3% |
| As | 0 | 0.0% | 25 | 100.0% |
| Sport | 0 | 0.0% | 28 | 100.0% |
| Mundo Deportivo | 1 | 3.6% | 27 | 96.4% |
| Total | 3 | 2.9% | 101 | 97.1% |

Thirdly, the existing link between Marca/As newspapers and news events related to one Real Madrid player is analyzed. Similarly, the existing link between Mundo Deportivo/Sport and news events related to one FC Barcelona player is analyzed. It is observed that Marca and As mainly focus their news about signings on Real Madrid (78.3% and 88%, respectively: Average 83.2%), while Mundo Deportivo and Sport do likewise with FC Barcelona (96.4% and 92.9%, respectively: Average 94.6%). There are only five pieces of news not related to either Real Madrid or FC Barcelona: Two about Atlético de Madrid (appearing in Marca) and three referring to other teams (two in As and one in Marca). Similarly, Mundo Deportivo and Sport did not mention any signing not referring either to Real Madrid or FC Barcelona on their front pages (Table 7). If we compare As-Marca with Mundo Deportivo-Sport, significant differences are found, especially in the number of stories devoted to the different football teams ($\chi^2$ = 78.21; $p$ < 0.001).

**Table 7.** Team affected by the signing, depending upon newspaper. Compiled by author.

| | Team Affected at Origin or Destination | | | | | | | | | |
| | Real Madrid | | Barcelona | | Atlético de Madrid | | Otros | | No Indicado | |
| | n | % | n | % | n | % | n | % | n | % |
| --- | --- | --- | --- | --- | --- | --- | --- | --- | --- | --- |
| Marca | 18 | 78.3 | 2 | 8.7 | 2 | 8.7 | | | 1 | 4.3 |
| As | 22 | 88.0 | 1 | 4.0 | | | 2 | 8.0 | | |
| Mundo Deportivo | 1 | 3.6 | 27 | 96.4 | | | | | | |
| Sport | 2 | 7.1 | 26 | 92.9 | | | | | | |
| Total | 43 | 41.3 | 56 | 53.8 | 2 | 1.9 | 2 | 1.9 | 1 | 1.0 |

Finally, 97.1% are accompanied by photographs. Marca accompanied all their news about signings with a photograph. In other newspapers, it can be noted that there is one piece of news in each with no photograph attached. This is not a significant difference ($\chi^2$ = 0.888; $p$ = 0.828) (Table 8).

**Table 8.** Use of photography, depending upon newspaper. Compiled by author.

| Newspaper | Photography | | | |
| | Yes | | No | |
| | n | % | n | % |
| --- | --- | --- | --- | --- |
| Marca | 23 | 100.0% | 0 | 0.0% |
| As | 24 | 96.0% | 1 | 4.0% |
| Sport | 27 | 96.4% | 1 | 3.6% |
| Mundo Deportivo | 27 | 96.4% | 1 | 3.6% |
| Total | 101 | 97.1% | 3 | 2.9% |

## 6. Conclusions

First of all, we assume the first hypothesis to be correct, since at least 50% of the signings, possible signings, renewals, or sale of players announced on the front pages did not materialize. This constitutes a general trend in sports journalism, especially during the summer. The summer transfer window is usually the perfect occasion for newspapers to speculate with rumors. That is so, precisely because of the lack of sports competitions, especially in the football field. That is why many of these pieces of news do not materialize in the end.

In this sense, from the perspective of the practice of journalism, especially sports press, rumor is a common way of journalists establishing authority. In addition, in this type of information, where there is logically no exclusivity (as for example in sports broadcasts), it is a strategy to maintain interest on the part of the reader, especially in the period studied (summer period) where the number of broadcasts is much lower than at any other time of the year. In addition, as other authors did in previous research [46], there are many occasions when sports media are followed by the "historically symbiotic relationship" that can be established between the journalist and their audience and, for example, in that same study, the lack of sources in the commercial forecast is contemplated.

Secondly, a minimal percentage of these front pages were accompanied by photo montage, so we reject the second hypothesis, which stated just the opposite premise. It is true that quite often there are photographs of the players (news events); however, they are usually related to the team that the player is part of, at that specific time.

On the other hand, there is a direct relationship between Marca/As and news events related to Real Madrid, and Mundo Deportivo/Sport and news events related to FC Barcelona. As a consequence of that, the third and fourth hypotheses are broadly accepted. Nevertheless, if a thorough analysis is carried out, we note that Marca does not reach the expected 80% stated in the hypothesis. All of this is in line with the editorial biases of the different newspapers and their own target groups. In this case, we can conclude that Marca publishes the widest range of news, including more news (not referring to Real Madrid) than was expected in the first place.

Finally, the fifth hypothesis is accepted, since photography is present in more than 80% of the front pages analyzed. In this aspect, there were not any significant differences among newspapers.

Future lines of research, or retort publications, could try to delve into whether the results and conclusions presented here could have significant differences if the whole of the news (text analysis) were analyzed beyond the headline of the cover; and if the title page would be a mere "clickbait" on a text, which would clarify what is specified there, in a less blunt and more speculative way (rumor).

**Author Contributions:** Conceptualization, F.-J.H.-G. and J.-D.U.-L.; methodology F.-J.H.-G. and J.-D.U.-L.; software, F.-J.H.-G. and J.-D.U.-L.; validation, J.-D.U.-L.; formal analysis, F.-J.H.-G. and J.-D.U.-L.; investigation, F.-J.H.-G. and J.-D.U.-L.; resources, F.-J.H.-G. and J.-D.U.-L.; data curation, J.-D.U.-L.; writing—original draft preparation, F.-J.H.-G.; writing—review and editing, F.-J.H.-G. and J.-D.U.-L.; supervision, F.-J.H.-G. project administration, F.-J.H.-G.; funding acquisition, F.-J.H.-G. All authors have read and agreed to the published version of the manuscript.

**Funding:** This research received no external funding.

**Informed Consent Statement:** Not applicable.

**Data Availability Statement:** Not applicable.

**Acknowledgments:** The authors thank the Department of Sociology and Communication at the University of Salamanca, for the financial support in the translation of this article.

**Conflicts of Interest:** The authors declare no conflict of interest.

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
