# Peer review of "The Importance of Rumors in the Spanish Sports Press: An Analysis of News about Signings Appearing in the Newspapers Marca, As, Mundo Deportivo and Sport"

_publications, doi:10.3390/publications9010009_

Round 1
Reviewer 1 Report
Thank you for the opportunity to review your paper, “The importance of rumors in the Spanish sports press.” The findings here do contribute to our knowledge of the Spanish press and sports media practice in the major national papers. The point about where the rumors appear do suggest distinctions in expectations about audiences, which is useful to know.
Fundamentally, I question whether fake news is the proper analytic framework for this research. The problem is best encapsulated in Line 145. Discussions about transfers that do not materialize may take place regularly. Porto could call Real Madrid about a player, be quoted a price that is too high and then abandon the effort for a moment. The transfer is unlikely, but it would be true that a conversation between the clubs occurred. Reporting that would not be fake news even if it would not be a likely transfer.
From a journalism practice perspective, the rumor is an important way to establish authority because journalists are insiders and find out things before they become public. The breaking of news is most possible in issues of transfer because, unlike in live broadcasts or game reports, reporters and outlets can have exclusivity over stories. The emphasis on rumor during the summer does seem to play into a strategy of keeping football journalists relevant during a period with few games. I might suggest deepening the analysis. In 2019, Reed and Harrison looked more closely at the sourcing in these sorts of stories. I might suggest an approach to whether the transfer stories cite sources or not.
Here is the citation for the article above: Reed, S., & Harrison, G. (2019). “Insider Dope” and NBA Trade Coverage: A Case Study on Unnamed Sourcing in Sport Journalism. International Journal of Sport Communication, 12(3), 419-430).
I am happy to review a revision, should the editors decide.
Author Response
First of all, we would like to thank for the comments offered by both evaluators, and then we go on to indicate the changes made to improve the article and to boost their final consideration for a publication:
1.-) In accordance with reviewer’s 1 remarks and agreeing with his observations, we have introduced a paragraph after line 152, in order to try to clarify the interpretation that we have given to the investigation:
“In order to understand the concept of fake news in any of these variants, research is based on the fact that, almost always, these news are given as true even though they are not, nor will they ever be, beyond the fact that there could be fond indications of veracity which may occur (for example, a transfer), either with a greater or lesser probability. The problem here is the lack of use of the conditional tense, which would surely detract from the ultimate goal of the headline. So it is preferable to start for example "Luis Suárez will sign for Atlético de Madrid" instead of "Luis Suárez could sign for Atlético de Madrid". Obviously, the use of the conditional indicates that "could" or "could not" and the news would cease to be such, when all options were left open”.
2.-) In agreement with several other remarks by reviewer 1, we have introduced a paragraph in the conclusions, after hypothesis 1, based on the correct indication offered, which complements the conclusion:
“In this sense, from the perspective of the practice of journalism, especially sports press, rumor is a common way of journalists establishing authority. In addition, in this type of information, where there is logically no exclusivity (as for example in sports broadcasts), it is a strategy to maintain interest on the part of the reader, especially in the period studied (summer period) where the number of broadcasts is much lower than at any other time of the year. Also, as other authors did in previous research [46], there are many occasions when sports media are followed by the “historically symbiotic relationship” that can be established between the journalist and their audience and, for example, in that same study, the lack of sources in the commercial forecast is contemplated”.
3.-) We also add the reference suggested by reviewer 1 in that same conclusion: "Insider Dope" and NBA Trade Coverage: A Case Study on Unnamed Sourcing in Sport Journalism. International Journal of Sport Communication, 12(3), 419-430). In conclusions, new line 332 and 247 (Methodology).
4.-) As stated by reviewer 2, methodologically we have tried to carry out a bibliographic review and we refer to the contributions found.
5.-) As stated by the reviewer 2, and given the difficulty at analyzing the full text of all the news within the 10 days we have to return the manuscript, in order to sort this out, we have added a future line of research at the end of the conclusions, also referring to the simile of clickbait pointed out by the reviewer:
“Future lines of research, or retort publications, could try to delve into whether the results and conclusions presented here could have significant differences if the whole of the news (text analysis) were analyzed beyond the headline of the cover; and if the title page would be a mere "clickbait" on a text which would clarify what is specified there, in a less blunt and more speculative way (rumor)”.
6.-) We have added the URLs to table 5.
7.- A paragraph has been added at the end of methodology to clarify the contrast of data with the official source of La Liga (https://www.laliga.com/); This URL has also been introduced in the bibliography:
“After conducting a systematic research through the main bibliographic reference sources (WOS, Scopus mainly), as well as recent review papers [42, 43, 44 and 45], we do not know of any other research papers similar to the analysis proposed here, with content methodology of a representative sample of news items that appeared in the press, in which the researchers themselves contrasted the truth of the news, with the exception of an isolated case taken to the United States sports arena [46]. Therefore, this methodological approach is novel and can be applied to any type of news, not only in the sports field.
The final veracity of the news was contrasted with knowing if a certain player was still part of the squad or not, through the official data from La Liga (https://www.laliga.com/) [47] taking into account that all players who appear on this headline would have a relationship with the Spanish League (either because they were signing for a Spanish club or were no longer part of the squad of a Spanish club)”.
8). Direct links to EGM have been added in Figures 1 and 2.
9.-) Bibliography on the content analysis technique has been updated and a citation of this technique used in another investigation to study fake news has been added.
10.-) Sources have been added to explain the importance of sports shows on television and the incidence of sports shows on radio. New references 3 to 7.
11.-) Direct URLs have been added to data from the four main sports newspapers, from the Alexa.com meter. New references 8 to 11.
12) Corrected in table 4 "fake news".
Hoping that the changes made may be sufficient to pass the evaluation process, we would like to convey our gratitude for your time and attention.
Looking forward to hearing back from you.
Yours sincerely, the authors.
Reviewer 2 Report
The topic covered in this publication is interesting. However, the methodology is the weakest part of this article. Thus in the section 4 (lines 215-225) when describing the methodology authors cited 3 works [29,30,31] which are relevant to the filed, but which not described recent approaches for such tasks (published in 1996, 1980 and 1952 respectively). If authors proposed novel approach, then I suggest to add more information about it and compare it to other similar works (including recent ones). For example, why authors didn’t considered full text analysis of the news, which are mentioned in the front page? How the front page looks like (examples, some URLs if possible)? Maybe “hot” or “controversial” title of the news can be used for “clickbait”, but details of the news will show something else? Which source of information (dataset) authors used to verify information (to decide if the content is fake or not)? Finally, is the method applicable for other languages and topics (not only sport)?
Other comments:
Figure 1 presents a ranking of the most read newspapers in Spain. Authors mentioned the source as "EGM 2020". The source [1] is the main page of the EGM website. Is there any direct link (to webpage or other publication) with such source data? The same is for Figure 2.
I suggest to add source(s) to statements which explain that the most watched TV or radio programs in Spain are related to sports (lines 52-59).
Sentence about importance of websites related to sports (lines 60-61) can be verified by relevant services such as alexa.com or similarweb.com. I suggest to enrich information about importance of the considered websites by using those or other popular tools.
In Table 4 “No (fake new)” -> “No (fake news)”
Author Response

(The authors gave the same response as above.)

Round 2
Reviewer 1 Report
The authors have addressed my concerns, and I am happy to recommend publication. I think these data could help set up a more in depth study as well.
Author Response
Thanks!
Reviewer 2 Report
Figure 2 must have a legend (so reader can understand which of the colors are related to particular type of media).
I suggest to provide additional information on the Internet sources (not only URL adresses) in References section .
In the abstract (line 17) I suggest to use "120 different issues of the newspaper" instead of "120 copies".
In line 248 authors wrote that "this methodological approach is novel and can be applied to any type of news". There are different studies on automatic detection of fake news or rumours. For example, in 2017 following well-cited works were published (with links to preprints):
- Automatic detection of fake news - https://arxiv.org/abs/1708.07104
- "liar, liar pants on fire": A new benchmark dataset for fake news detection - https://arxiv.org/abs/1705.00648
- Simple open stance classification for rumour analysis - https://arxiv.org/abs/1708.05286
The above works proposed approaches, which can automatically detect fake news (or rumours) and can be used on different sources/media. The question is - why authors can not use such solutions in this study? If it is not applicable, I suggest to explain why is so and what are the advantages to use own approach.
Author Response
Thanks to the reviewer.
We have added to figure 2 a legend. Now, readers can understand which of the colors are related to particular type of media. Morever, we have added another media, not only three.
We have provided additional information on the Internet sources (not only URL adresses) in references section: 4 to 11 and “new” 50 (cyan color).
In the abstract (line 17) new text: "120 different issues of the newspaper" instead of "120 copies" (cyan color).
In lines 248-251, we have added articles suggested by the reviewer:
- Pérez-Rosas, V., Kleinberg, B., Lefevre, A. and Mihalcea, R. (2018). Automatic Detection of Fake News. Proceedings of the 27th International Conference on Computational Linguistics, pages 3391–3401. https://www.aclweb.org/anthology/C18-1287.pdf
- Yang Wang, W. (2017). "Liar, Liar Pants on Fire": A New Benchmark Dataset for Fake News Detection. Proceedings of the 55th Annual Meeting of the Association for Computational Linguistics (Short Papers), pages 422–426 Vancouver, Canada, July 30 - August 4, 2017. In: https://www.aclweb.org/anthology/P17-2067.pdf DOI: https://doi.org/10.18653/v1/P17-2067
- Aker, A. Derczynski, L. and Bontcheva, K. (2018). Simple Open Stance Classification for Rumour Analysis. Proceedings of Recent Advances in Natural Language Processing, pages 31–39, Varna, Bulgaria, Sep 4–6 2017. In: Proceedings of Recent Advances in Natural Language Processing, pages 31–39, Varna, Bulgaria, Sep 4–6 2017. https://doi.org/10.26615/978-954-452-049-6_005 https://doi.org/10.26615/978-954-452-049-6_005
and we have tried to justify our study by not being from online newspapers or social networks but traditiconal newspapers.
We hope the new version can be right.